# Implementation of an open-source robotic platform for SARS-CoV-2 testing by real-time RT-PCR

José Luis Villanueva-Cañas[1]*, Eva Gonzalez-Roca[1,2], Aitor Gastaminza Unanue[3], Esther Titos[1,4,5], Miguel Julián Martínez Yoldi[1,6], Andrea Vergara Gómez[1,6], Joan Anton Puig-Butillé[1]

1 Molecular Biology CORE (CDB), Hospital Clínic de Barcelona, Barcelona, Spain, 2 Immunology Department (CDB), Hospital Clínic de Barcelona, Barcelona, Spain, 3 Independent engineer, Barcelona, Spain, 4 Department of Biochemistry and Molecular Genetics (CDB), Hospital Clínic de Barcelona, Barcelona, Spain, 5 Department of Biomedical Sciences, University of Barcelona, Barcelona, Spain, 6 Department of Microbiology (CDB), Hospital Clínic de Barcelona, Barcelona, Spain

* jlvillanueva@clinic.cat

**Data Availability Statement:** Code developed is available at https://github.com/CDB-coreBM/ covid19clinic.

## Abstract

The current global pandemic due to the SARS-CoV-2 has pushed the limits of global health systems across all aspects of clinical care, including laboratory diagnostics. Supply chain disruptions and rapidly-shifting markets have resulted in flash-scarcity of commercial laboratory reagents; this has motivated health care providers to search for alternative workflows to cope with the international increase in demand for SARS-CoV-2 testing. The aim of this study is to present a reproducible workflow for real time RT-PCR SARS-CoV-2 testing using OT-2 open-source liquid-handling robots (Opentrons, NY). We have developed a framework that includes a code template which is helpful for building different stand-alone robotic stations, capable of performing specific protocols. Such stations can be combined together to create a complex multi-stage workflow, from sample setup to real time RT-PCR. Using our open-source code, it is easy to create new stations or workflows from scratch, adapt existing templates to update the experimental protocols, or to fine-tune the code to fit specific needs.

Using this framework, we developed the code for two different workflows and evaluated them using external quality assessment (EQA) samples from the European Molecular Genetics Quality Network (EMQN). The affordability of this platform makes automated SARS-CoV-2 PCR testing accessible for most laboratories and hospitals with qualified bioinformatics personnel. This platform also allows for flexibility, as it is not dependent on any specific commercial kit, and thus it can be quickly adapted to protocol changes, reagent, consumable shortages, or any other temporary material constraints.

## Introduction

The coronavirus disease 2019 (COVID-19) pandemic, caused by the severe acute respiratory syndrome coronavirus (SARS-CoV-2) [1], continues to have a devastating impact worldwide. Testing for COVID-19 is essential to control and monitor the virus transmission, and to

**Funding:** The author(s) received no specific funding for this work.

**Competing interests:** The authors have declared that no competing interests exist.

manage health resources [2]. Although other COVID-19 testing methods are available, the real time Reverse Transcription Polymerase Chain Reaction (RT-PCR) remains the gold-standard and the main molecular approach [3]. However, in the hardest-hit countries, the shortage of laboratory supplies and limited testing capacity during the first months of the pandemic resulted in a restriction in the use of RT-PCR to few population groups (e.g. patients with severe symptoms, high-risk patients, or healthcare workers) [4, 5]. Hospitals and laboratories were forced to incorporate automated solutions, and simultaneously, to increase the through-put of existing ones [6].

There are several commercial platforms for automated high-throughput testing, such as the Cobas 6800 system from Roche [7] or the Panther Fusion system from Hologic [8]. These solutions usually work with proprietary reagent cartridges specifically tailored to each platform and process, using little or no generic components or consumables. This limits the utility of such platforms if the supply chains of a specific labware or reagent fail. Other platforms, such as liquid handling automated workstations from companies such as TECAN or Hamilton are high throughput platforms that can accommodate different protocols. In response to COVID-19, these companies adapted extraction, qPCR, and ELISA protocols to run on their platforms [9, 10]. These platforms are programmed through their proprietary software, and users are able to create their own scripts to some extent. Although they are relatively flexible, their high price is still a barrier to many laboratories.

The advent of affordable open-source liquid handling robots, such as the OT-2 developed by Opentrons, opened the door to low-cost automated solutions [11]. Despite their limitations, multiple OT-2 units can be used together to produce more complex workflows which would otherwise be impossible to do with a single OT-2 robot. Furthermore, as they are relatively affordable and open-source at both hardware and software levels, this makes them an attractive alternative for many laboratories due to the fluctuating demands of the health care market during a pandemic [12]. The "open-everything" approach allows these machines to circumvent supply chains disruptions of specific laboratory equipment vendors, as OT-2 robots can be easily configured to work with a wide variety of components and consumables. Open software is another advantage, as new protocols and features can be quickly be shared with other users. In addition, OT-2 robots are programmed using Python, which is the third most used programming language worldwide with an estimated community of 8.5 million developers [13].

In April 2020, the Hospital Clínic de Barcelona, a tertiary hospital in Spain, incorporated a set of OT-2 robots to increase the RT-PCR SARS-CoV-2 testing capacity.

In this work, we first describe the framework we developed to implement RT-PCR SARS-CoV-2 testing using the OT-2 robots at our center. This framework includes a structured code template with built-in functions that help to overcome some of the robots' limitations.

Secondly, using our framework, we designed a "circuit" for SARS-CoV-2 RT-PCR testing, by combining several OT-2 robots which each perform a different stage of the test protocol. Finally, we evaluated our circuit using samples from an external quality assessment (EQA) and compared it to other routine automated diagnostic platforms.

## Methods

### Developing the framework

The OT-2 robots were initially created to perform simple pipetting tasks in research laboratories, and have a limited number of actions compared to other commercial liquid handlers due to their small size. After deploying, calibrating, and testing the robots, we detected some limitations with regard to incorporating the robots into a clinical setting. First, we realized that we needed to develop a framework to ameliorate hardware limitations but which could also help

us cope with changes in labware, number of samples, or reactive volumes. We also needed to simplify the process of creating or modifying stations, and automate the circuit as much as possible.

We developed code that provides features to improve the base capabilities of the OT-2 robots and can be used as a starting point to develop individual stations and complete circuits. The Python code (functions and classes) is documented in our GitHub wiki: https://github.com/CDB-coreBM/covid19clinic/wiki. It includes general descriptions of the functions and examples.

We also created a general template that includes all the functions developed and that can be used to facilitate the development of new stations as well as the automation of some tasks. The template has been designed to work with a variable number of samples and includes the features described below. Furthermore, the template is structured in distinct steps, making the code easier to understand and repurpose. This structure allows the user to execute selected steps, which can be useful for testing or resuming a run after encountering a problem. Some improvements and functions include:

**Reagent classes.** The liquids used in a protocol might have distinct physical properties. We developed a system that defines liquids as python objects with predefined properties at the beginning of a protocol. With this system, it is only necessary to specify which liquid is being used in a step to apply all the adjustments for that liquid automatically. For example, when dispensing alcohol-based substances, a coating of the liquid is left on the tip; in these scenarios, the attribute "rinse" can be enabled. Pre-rinse is a technique to increase pipetting accuracy through the pre-conditioning of tips by aspirating a certain volume prior to using the tip. A more detailed description of this function is available on GitHub.

**Height adjustments (function).** The reagent classes store the total volumes, location, and remaining volumes of all liquids defined. By using this information, combined with the geometrical measures of the containers, we used a simple formula to calculate the optimal aspiration height when pipetting that liquid.

$$Pickup\ height = Total height - aspirated\ height$$

$$Total\ height = x\frac{(Container\ volume - cone\ volume)}{container\ cross\ section\ area} + \frac{3x\ cone\ volume*}{cone\ base\ area}$$

$$aspirated height = \frac{aspirated\ volume}{container\ cross\ section\ area}$$

*Formula for a container with cone-shaped bottom. The formula should be adjusted for different shapes. We include a thorough description and simulations in S1 File.

*Other enhanced functions*. These include a highly customizable liquid transfer function, which makes use of the reagent classes to adjust different parameters, a multi-dispensing function for one to many transfers or a reagent mixing function, among others.

**Log system.** Currently, every run generates a time report, registering the duration of each step and the number of pipette tips used.

**Automation.** An essential aspect of implementing a circuit in clinical diagnostics is the traceability of samples. Sample tracking is very context-dependent, as each laboratory has its own protocols, laboratory information management systems (LIMS), and databases. We provide several scripts that can be adapted to different working environments; for example, a template for sample barcode scanning that tracks sample placement in the different possible dispositions (i.e. 4 racks with 24 samples each or qPCR with 96 wells), or the automatic

generation of templates with sample names and well positions for the ABI 7500 qPCR system (Thermo Fisher Scientific) or the LightCycler 480 II system (Roche life science).

**Custom labware.** Definitions of commonly used plasticware from popular vendors are available for download from our repository. These are available as.json files, with the measures and their description, and can be imported into the OT-2 app.

## Circuit design and evaluation

Using our framework, we build two circuits for SARS-CoV-2 RT-PCR testing using OT-2 robots. A full description of these circuits is provided in the Results section. The difference between both circuits relies on the kit used for RNA extraction: MagMax™ Pathogen RNA/DNA kit or MagMax™ Viral Pathogen II RNA/DNA kit. Both circuits use TaqPath™ COVID-19 CE-IVD RT-PCR reagents for the RT-PCR reaction.

The first circuit was named OT-2KF pathogen, after the extraction kit used (MagMax™ Pathogen RNA/DNA kit). We externally evaluated this circuit using eight inactivated samples with unknown content (at the time of processing) provided by the European Molecular Genetics Quality Network (EMQN). These samples were analyzed using the KF pathogen circuit and three additional platforms used in routine at our laboratory: Flow Solution (Roche), Cobas 6800 (Roche), and a Hamilton-Seegene combination (see S1 File). Samples were mixed 1:1 with Cobas omni Lysis Reagent (Roche) before being processed to allow comparisons.

Due to shortages in MagMax™ Pathogen RNA/DNA kit, we had to build a second circuit (OT2-KF viral pathogen II) using MagMax™ Viral Pathogen II RNA/DNA extraction kit. To evaluate our second circuit, we compared it to the OT-2-KF pathogen circuit using 15 previously collected nasopharyngeal swab samples (with known outcome) from our center. These samples were collected for routine diagnostics and immediately placed into a sterile tube, containing 2 mL of lysis buffer (2 M guanidinium thiocyanate, 2 mM dithiothreitol, 30 mM sodium citrate, and 1% Triton X-100). Samples were vortexed and mixed 1:1 with Cobas omni Lysis Reagent (Roche, Basel, Switzerland) for inactivation.

**Nucleic acid extraction and SARS-CoV-2 qPCR set up.** Viral RNA was extracted from inactivated samples using either the MagMax™ Pathogen RNA/DNA kit or the MagMax™ Viral/Pathogen II kit on the Kingfisher™ Flex Purification System (Thermo Fisher Scientific, Waltham, MA) according to manufacturer's instructions with some modifications. Specifically, when we used the MagMax™ Pathogen RNA/DNA kit protocol, no addition of MagMax™ Pathogen RNA/DNA Lysis Solution Master Mix was applied to the inactivated samples, and 460 µl of lysates were directly transferred to the King Fisher™ deep-well sample plate. Subsequently, 260 µl of MagMax™ Pathogen RNA/DNA Bead Master Mix and 5 µl of the MS2 Phage Control (Thermo Fisher Scientific, Waltham, MA) were added to each well, including a negative control sample containing 460 µl of PBS located at A1 position of the 96 deep-well plate. The sample plate was then loaded onto the KingFisher™ instrument, and automated RNA extraction was performed using the BindIt™ protocol "MagMax_Pathogen_High_Volume".

Viral RNA extraction using the MagMax™ Viral/Pathogen II kit was performed according to the protocol for "400 µl sample input with two wash steps" recently published by Thermo Fisher Scientific as a "Sample Prep Application Note" on its website with some modifications. To summarize the protocol, no addition of proteinase K was applied to the inactivated samples and 400 µl of sample were directly transferred to the King Fisher™ deep-well sample plate. Subsequently, 550 µl of Binding Bead Mix and 10 µl of the MS2 Phage Control (Thermo Fisher Scientific, Waltham, MA) were added to each well including a negative control sample containing 400 µl of PBS located at A1 position of the 96 deep-well plate. Sample plate was then loaded onto the KingFisher™ instrument and, automated RNA extraction was performed using

the BindIt™ protocol "MVP_2Wash_400_Flex" provided in Thermo Fisher's website with minor modifications. Specifically, the proteinase K digestion step was removed and plastic configuration was used as follows: three King Fisher™ deep-well plates (samples, wash1 and wash2), 2 King Fisher™ Standard plates (elution and Tip Comb plate) and one Tip Comb. See Availability Section for more information.

**RT-PCR in OT-2/Kingfisher circuits.** To summarize the protocol, 5 μl of nucleic acid eluate was used for multiplex real-time RT-PCR test intended for the qualitative detection of nucleic acid for SARS-CoV-2 using the TaqPath™ COVID-19 CE-IVD RT-PCR kit (Thermo-Fisher Scientific, Waltham, MA).

**Interpretation of SARS-CoV-2 qPCR results.** Results from the ABI 7500 qPCR system, generated by the 7500 software v2.3.3 (Thermo Fisher Scientific), are transformed into a user-friendly table using an Rmarkdown script, which includes an automatic interpretation of results. A sample is considered positive when two or more targets are amplified, negative when the internal control is amplified but the viral targets are not, or invalid when there is no ampli-fication of any targets (including internal control). Other cases are marked as "revise" and eval-uated individually by a microbiologist prior to automatically transferring the results to the laboratory information system.

## Results

### Building circuits with OT-2 robots

We have built and tested two different functional circuits for SARS-CoV-2 RT PCR testing using our code framework in the OT-2 stations. Both circuits work with equipment and reagents from Thermo Fisher Scientific, which are routinely used for SARS-CoV-2 testing in our laboratory and are already validated. Each circuit is composed of six stations: four OT-2 stations, a Kingfisher™ Flex Purification System, and one ABI 7500 Fast Real-time PCR instru-ment. As previously mentioned, the difference between both circuits relies on the kit used for RNA extraction (see Methods).

In both cases, the circuit workflow (Fig 1A) is as follows:

1. Initial sample setup (OT-2 station A), 2. sample preparation (OT-2 station B1), 3. plate filling for Kingfisher Flex (OT-2 station B2), 4. RNA extraction step (Kingfisher Flex station), 5. RT-PCR preparation (OT-2 station C), and the 6. RT-PCR (ABI 7500 Fast thermocycler).

Each OT-2 station can have two pipettes simultaneously installed. The OT-2 pipettes have three different capacities (20, 300, 1000 μl) and can be either single-channel or multi-channel. A combination of these pipettes defines letters in the OT-2 station nomenclature; therefore sta-tions B1 and B2 can be run on the same robot. A complete description of every station with its required list of components is available as S2 File.

The complete process for testing 96 samples takes about 4h (Table 1), and a single labora-tory technician can operate it. RT-PCR amplification and detection using ABI 7500 Fast ther-mocycler is the longest step. Theoretically, a new run could start every 70 minutes. A summary of the steps carried out in each station and their executing time is indicated in Fig 2.

### Evaluation of OT-2-KF pathogen circuit within a clinical setting

We analyzed the samples (N = 8) from the Coronavirus Outbreak Preparedness EQA Pilot study from EQMN using the OT-KF pathogen circuit. We found that, out of eight samples, five were positive for SARS-CoV-2, and three were negative. The results obtained by the OT-KF pathogen circuit were consistent with those obtained by three commercial platforms, even though the target regions were different (Table 2). The internal control was correctly amplified for all samples and platforms.

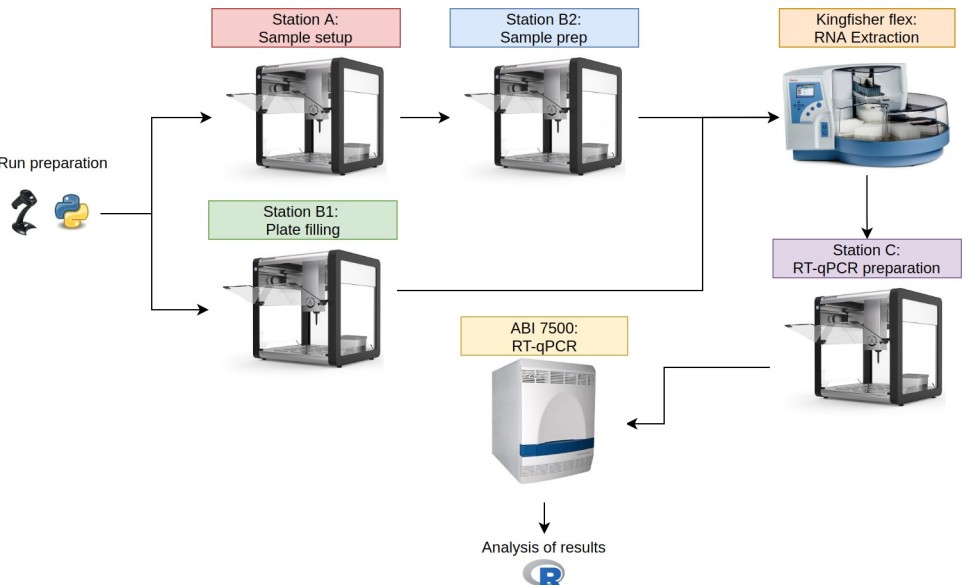

**Fig 1. Diagram of the OT2-KF circuit.** Note that station A and station B1 can be run simultaneously. The full configuration of labware for OT2 stations is available as S2 File.

Ct levels are inversely proportional to the amount of target nucleic acid in a sample. The range of Ct was 25.8–35.25 for Flow Solution platform, 25.47–35.82 for Cobas 6800, and 27.72–36.91 for Hamilton-Seegene. The range of Ct for the platform OT2-KF pathogen was 23.66–38.02, with consistently lower values except for the S gene in sample CVOP2052-08.

According to the EMQN sample content, we identified all the SARS-CoV-2 strains correctly (Table 3) with all platforms. The other coronaviruses (NL63 and OC43) were not detected, showing that the tests have specificity for SARS-CoV-2.

## Comparison of OT-2-KF pathogen circuit with OT2-KF viral pathogen II

Fifteen previously frozen samples with known outcome were run across the two KF circuits in two separate batches. We obtained identical qualitative results for all samples in both circuits (Table 4). Sample S8 had one amplified target (N–nucleocapsid gene) with a very high Ct value (37.9), compatible with a patient after infection with some remnants of the virus.

The Ct values for the OT-2-KF viral pathogen II were 1.6 higher on average. One possible explanation for this difference is that the samples had an additional round of freezing-unfreezing as they were processed in the second batch.

**Table 1. Number of steps and running times in minutes for a full run with 96 samples across the different stations.** Station A and B1 can be run simultaneously.

| Station | Steps | Total time (minutes) |
|---|---|---|
| A: sample setup | 1 | 54 |
| B1: plate filling | 5 | 41 |
| B2: sample prep | 3 | 59 |
| C: RT-PCR preparation | 2 | 20 |
| KF: extraction | 1 | 30 |
| Thermocycler: RT-PCR | 1 | 70 |
| **Total** | | 234 |

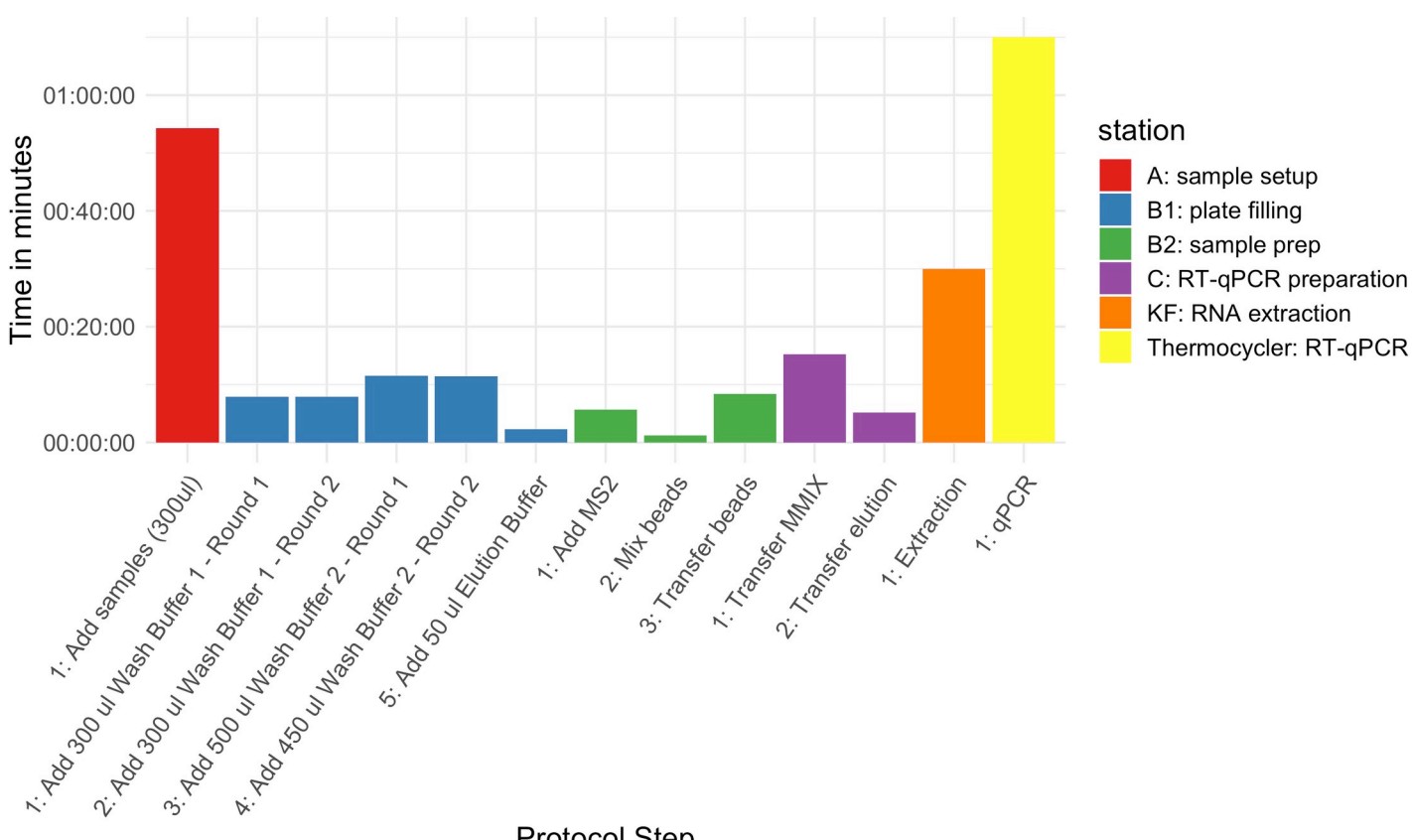

**Fig 2. Run times in minutes for the different steps classified by station in the Kingfisher pathogen circuit using the MagMax™ Pathogen RNA/DNA kit.**

## Discussion

We have successfully prototyped and implemented an automated robotic workflow for performing SARS-CoV-2 RT-PCR tests using OT-2 robots and other commercial systems, as well as validating its viability at the production level. We obtained results which were equivalent to other commercial automated solutions, valid results according to the Coronavirus Outbreak Preparedness EQA Pilot study EMQN scheme, indicating that our custom framework can be routinely used in a clinical setting. So far, we have processed over 40,000 samples in more than

**Table 2. Comparison of four different platforms used for SARS-CoV-2 detection from external quality control samples, including our OT2-KF circuit.** Every platform has its own targets and primers.

| | Flow Solution | Cobas 6800 | | Hamilton-Seegene | | | OT2-KF pathogen | | |
|---|---|---|---|---|---|---|---|---|---|
| Sample ID | E | E | ORF1 | E | N | RdRP | N | ORF1ab | S |
| **CVOP2052-01** | 29,57 | 30,43 | 29,74 | 28.19 | 30.81 | 31.26 | 28,07 | 27,3 | 28,15 |
| **CVOP2052-02** | Undetectable | Undetectable | Undetectable | Undetectable | Undetectable | Undetectable | Undetectable | Undetectable | Undetectable |
| **CVOP2052-03** | 32,15 | 33,8 | 32,7 | 31.73 | 33.39 | 34.43 | 31,08 | 30,35 | 31,16 |
| **CVOP2052-04** | Undetectable | Undetectable | Undetectable | Undetectable | Undetectable | Undetectable | Undetectable | Undetectable | Undetectable |
| **CVOP2052-05** | Undetectable | Undetectable | Undetectable | Undetectable | Undetectable | Undetectable | Undetectable | Undetectable | Undetectable |
| **CVOP2052-06** | 29,06 | 31,14 | 30,23 | 28.93 | 30.94 | 31.31 | 27,3 | 26,92 | 27,81 |
| **CVOP2052-07** | 25,8 | 27,16 | 26,63 | 25.47 | 27.72 | 27.73 | 24,64 | 23,66 | 24,46 |
| **CVOP2052-08** | 35,25 | 35,82 | 35,27 | 34.49 | 35.98 | 36.91 | 34,42 | 34,42 | 38,02 |

**Table 3. Sample content of the EQA Pilot study.** *Sample code*: QCMD panel sample codes for the samples distributed to laboratories. *Sample content*: Content of the individual panel samples and, where applicable, the subtype or stain of the pathogen. *Expected result*: Expected outcome when testing for SARS-CoV-2. *Log10 dPCR Copies/ml*: The value obtained using a digital droplet PCR assay (modified from [14]). Samples CVOP20S2-07, 01, 03, 08, are in a calibrated dilution series. CVOP20S2-06 is a duplicate sample of CVOP20S2-01.

| Sample code | Sample content | Expected result | Log10 dPCR Copies/ml |
|---|---|---|---|
| CVOP20S2-01 | Novel Coronavirus SARS-CoV-2 | Positive | 4.30 |
| CVOP20S2-02 | Coronavirus NL63 | Negative | 4.64 |
| CVOP20S2-03 | Novel Coronavirus SARS-CoV-2 | Positive | 3.30 |
| CVOP20S2-04 | Coronavirus OC43 | Negative | 4.03 |
| CVOP20S2-05 | CV Negative | Negative | - |
| CVOP20S2-06 | Novel Coronavirus SARS-CoV-2 | Positive | 4.30 |
| CVOP20S2-07 | Novel Coronavirus SARS-CoV-2 | Positive | 5.30 |
| CVOP20S2-08 | Novel Coronavirus SARS-CoV-2 | Borderline Positive | 2.30 |

500 runs, and our code has been used as a starting point to develop SARS-CoV-2 OT-2 circuits in several other hospitals in Spain.

The framework includes classes and functions that facilitate the use of OT-2 liquid handling robots. For example, by creating a new class for different types of liquids, we have added a functionality that matches the more expensive commercial liquid handler robots available on the market. All attributes that depend exclusively on liquid properties are described in the code, and can be modified in the object definition section.

An advantage of establishing circuits based on independent modules is that some processes can be run in parallel, resulting in an increased turnaround and increased testing capacity. Taking into account the station times in the OT-2-KF circuit (Fig 1), the longest step is the RT-PCR amplification and detection using ABI 7500 Fast thermocycler. Therefore, while a run could theoretically start every 70 minutes, the inactivation of samples remains the main bottleneck. Inactivation of samples is a non-automated procedure that requires specialized facilities and microbiology laboratory technicians.

**Table 4. Comparison of fifteen samples using the two automated circuits implemented in our lab using Thermo Fisher kits: OT2-KF pathogen and OT2-KF Viral Pathogen II.**

| Sample | Kit Pathogen | | | | Kit Viral Pathogen II | | | |
|---|---|---|---|---|---|---|---|---|
| | N | ORF | S | MS2 | N | ORF | S | MS2 |
| S1 | Undetectable | Undetectable | Undetectable | 26,4 | Undetectable | Undetectable | Undetectable | 26,5 |
| S2 | 22,9 | 23,1 | 22,7 | 22,9 | 23,9 | 23,4 | 22,8 | 27,1 |
| S3 | 27,3 | 27,9 | 27,8 | 26,7 | 29,2 | 29,2 | 28,7 | 27,5 |
| S4 | 31,4 | 31,3 | 31,6 | 26,4 | 32,5 | 34,5 | 33,0 | 27,4 |
| S5 | 28,9 | 30 | 29 | 26,7 | 32,7 | 34,7 | 34,1 | 25,6 |
| S6 | Undetectable | Undetectable | Undetectable | 26,3 | Undetectable | Undetectable | Undetectable | 27,5 |
| S7 | 31,9 | 31,4 | 29,7 | 26,8 | 33,4 | 33,3 | 32,3 | 28,7 |
| S8 | Undetectable | Undetectable | Undetectable | 25,9 | 37,9 | Undetectable | Undetectable | 28,2 |
| S9 | 17,9 | 18,5 | 18,6 | 26,3 | 20,0 | 20,2 | 19,8 | 28,0 |
| S10 | 26,3 | 26,8 | 24,8 | 26,7 | 27,0 | 27,2 | 26,9 | 27,0 |
| S11 | 21,8 | 21,9 | 21,9 | 26,9 | 22,5 | 22,7 | 22,3 | 25,1 |
| S12 | 25,7 | 26,2 | 24,9 | 27,3 | 26,9 | 26,8 | 26,2 | 27,8 |
| S13 | 23,4 | 23,4 | 23,2 | 26,5 | 25,0 | 25,1 | 24,8 | 27,1 |
| S14 | 27,8 | 27,9 | 27,5 | 25,7 | 29,3 | 28,7 | 28,4 | 26,8 |
| S15 | Undetectable | Undetectable | Undetectable | 25,9 | Undetectable | Undetectable | Undetectable | 27,0 |

OT-2-based circuits are a promising solution to increase SARS-CoV-2 testing capability since it is an affordable open-source platform for liquid handling. The cost of the robots is much lower than other solutions, and the cost of reagents can be significantly reduced by using in-house protocols [15]. However, other associated costs (personnel, labware, etc.) are similar. Open-source platforms are also versatile, as it is easy to incorporate new protocols, adjust existing protocols to compensate for fluctuations in consumable availability, and benefit from code shared and maintained by the open-source research community. Although working with these platforms can be challenging at first, further improvements made by members of the community can be quickly shared and implemented across laboratories worldwide. Currently, the OT-2 community is still growing, and the available code is not always easy to repurpose; however, our work contributes code to the community that is flexible and scalable, and mimics the advanced capabilities available in more expensive proprietary platforms.

As previously mentioned, OT-2 circuits also have limitations. For instance, some protocols like RNA extraction are difficult or impossible to perform on the OT-2 platform. To date, most extraction protocols use magnetic beads and these protocols require some shaking steps. While OT-2 systems have a magnetic module, they currently do not incorporate an agitator. Despite this deficiency, we tested different RNA extraction protocols using OT-2 robots with partial success (data not shown). Therefore, in the design of our circuits where test accuracy and sensitivity were key, RNA extraction was carried out by a KingFisher™ instrument which incorporates an agitator module. However, advances and improvements to the OT-2 platform's capabilities continue, as evidenced by a new in-house extraction protocol which has been recently implemented using OT-2 robots in a hospital [15].

Another limitation to the OT-2 is that there is currently no hardware or software which performs sample tracking, which is essential in a clinical setting. We used external barcode readers before starting a run to associate individual samples to a pre-defined position on each plate, with a series of scripts for technicians that help to automate the process. Thirdly, OT-2 robots lack hardware to control if samples are correctly aspirated and/or detect clot formations. To minimize the clinical of impact of that limitation, OT-2 robots are monitored by a technician who visually inspects the plates between each station. By visually inspecting samples before testing, the majority of issues regarding clots or low volume sample are quickly detected. Problematic samples can be then separated from the rest and addressed on an individual basis. Additionally, pipetting is adapted to each liquid type by using reagent classes, as mentioned previously. For example, slow aspiration and dispense speed is set when pipetting SARS-CoV-2 samples to compensate for their high density and prevent clotting.

Another issue is that developing these circuits from scratch requires both in-house programming and molecular biology expertise. Even if general templates and working code are available, it will still be necessary to make changes, adjustments, and to understand the biological context. In that sense, the role of bioinformatics professionals in molecular diagnostics laboratories is fundamental when implementing such circuits.

In conclusion, this work can help to increase SARS-CoV-2 testing capacity of established laboratories and provide an alternative to proprietary protocols and during acute consumable shortages [15, 16]. These open-source solutions might also be a good fit for laboratories that currently do not have the resources for other automated platforms. Finally, we have developed a framework that can be used to adapt the OT-2 robots to other uses.

## Availability

Code, documentation, and station diagram and descriptions are available in the CDB GitHub repository: https://github.com/CDB-coreBM/covid19clinic

Codes for Kingfisher robot are available in Thermo Fisher's website:

Pathogen. https://www.thermofisher.com/order/catalog/product/4462359#/4462359

Viral Pathogen II. https://www.thermofisher.com/order/catalog/product/A48383#/A48383

## Supporting information

**S1 File. Supplementary methods.** Expanded methodology for the height calculation function and description of the platforms used in the EMQN comparison.
(DOCX)

**S2 File. Supplementary material.** Description of OT-2 stations including the labware placement for every circuit described in this manuscript.
(PDF)

**S1 Fig.**
(TIF)

## Acknowledgments

We thank M.D. Jiménez, Anabel Martínez, Mar López, Marta Parera, Núria Palau, Víctor Pastor, and Paula Sánchez for laboratory support and advice. Elena Roel, Katie Miller, and William Blevins for insightful comments. Ojas Patel for technical support. The authors thank the NGO COVIDWarriors for donating the Opentrons OT-2 stations to the Hospital Clínic de Barcelona (Spain).

## Author Contributions

**Conceptualization:** José Luis Villanueva-Cañas, Eva Gonzalez-Roca, Aitor Gastaminza Unanue.

**Formal analysis:** José Luis Villanueva-Cañas.

**Investigation:** Miguel Julián Martínez Yoldi.

**Methodology:** José Luis Villanueva-Cañas, Eva Gonzalez-Roca, Aitor Gastaminza Unanue, Esther Titos, Miguel Julián Martínez Yoldi, Andrea Vergara Gómez.

**Project administration:** Joan Anton Puig-Butillé.

**Resources:** José Luis Villanueva-Cañas, Esther Titos, Andrea Vergara Gómez.

**Software:** José Luis Villanueva-Cañas, Eva Gonzalez-Roca, Aitor Gastaminza Unanue.

**Supervision:** José Luis Villanueva-Cañas, Joan Anton Puig-Butillé.

**Validation:** José Luis Villanueva-Cañas, Esther Titos, Miguel Julián Martínez Yoldi.

**Writing – original draft:** José Luis Villanueva-Cañas.

**Writing – review & editing:** José Luis Villanueva-Cañas, Eva Gonzalez-Roca, Esther Titos, Miguel Julián Martínez Yoldi, Andrea Vergara Gómez, Joan Anton Puig-Butillé.

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
