## [Decision Letter · Decision Letter 0]

2 Mar 2021

PONE-D-21-03093

ROBOCOV: An affordable open-source robotic platform for SARS-CoV-2 testing by RT-qPCR

PLOS ONE

Dear Dr. Villanueva-Cañas,

Thank you for submitting your manuscript to PLOS ONE. After careful consideration, we feel that it has merit but does not fully meet PLOS ONE’s publication criteria as it currently stands. Therefore, we invite you to submit a revised version of the manuscript that addresses the points raised during the review process.

Authors should take into account all the recommendations of reviewers necessary for the publication of this work. Only in this case, the manuscript will be considered for publication.

We look forward to receiving your revised manuscript.

Kind regards,

Ruslan Kalendar, PhD

Academic Editor

PLOS ONE

Journal Requirements:

2. Thank you for stating the following after the Acknowledgments Section of your manuscript:

'FUNDING

This work was supported by the Hospital Clínic Barcelona'

'The author(s) received no specific funding for this work.'

Reviewers' comments:

Reviewer's Responses to Questions

**Comments to the Author**

1. Is the manuscript technically sound, and do the data support the conclusions?

Reviewer #1: Partly

Reviewer #2: No

2. Has the statistical analysis been performed appropriately and rigorously? 

Reviewer #1: No

Reviewer #2: No

3. Have the authors made all data underlying the findings in their manuscript fully available?

Reviewer #1: Yes

Reviewer #2: Yes

4. Is the manuscript presented in an intelligible fashion and written in standard English?

Reviewer #1: No

Reviewer #2: No

5. Review Comments to the Author

Reviewer #1: 

The title of the manuscript should be revised to be more informative, and, unless used with permission, to avoid the use of the name ROBOCOV, which is the name of a pre-existing statistical package – see https://github.com/kkdey/Robocov

As noted by the authors, the COVID-19 pandemic has highlighted the disastrous impact of supply chain disruptions on global response. Identification of cost-effective alternatives to increase diagnostic capabilities is critical. The authors describe integration of a potentially interesting robotic platform from a relatively new vendor, that has the advantage of utilizing open-source software, and compare its performance to several better-known systems that have FDA emergency use authorization for SARS-CoV-2 diagnostic testing. However, the manuscript can be shortened and would benefit from being more focused. In addition, the use of advertising jargon, eg. "Hit the ground running", or descriptions of system advantages that are not referenced or specifically supported by data in the manuscript may appear promotional and detract from the authors’ message, and should be deleted.

The authors’ stated aim is to describe use of the Opentrons OT system to set-up a reproducible workflow for diagnostic testing. It should be more clearly stated that the authors are describing a potential workflow incorporating the OT robot and the authors’ COVID-19 protocol for the Opentrons robot/author’s modifications to the COVID-19 protocol from the Opentrons Protocol Library, rather than a pilot study or an implementation in a clinical diagnostic setting, which would require testing of additional performance parameters and a larger sample size.

Additionally, since the authors are proposing this system as an affordable system for clinical diagnosis when resources may be limited, it would be helpful to discuss further:

System Affordability

- When the authors describe the system as affordable relative to other systems, are they considering only the cost of the robot itself? If so, it should be clearly stated that other major set-up costs, maintenance, consumables, personnel costs for operation etc. needed to obtain an interpretable diagnostic result for clinical purposes have not been addressed or compared to other systems

Consumables supply chain

- The authors raise concerns regarding global supply chain problems as a reason for using this open-source software platform. Please clarify why availability of the proprietary components and supplies listed on the Opentrons website, including heating blocks, pipette tips, racks & adapters, would not be an issue for this vendor.

Potential clinical impact of system limitations

- The authors note that there is no hardware or software in the robots to enable sample tracking. The value of robotics comes when large numbers of samples are being tested, and manual front and back end specimen tracking becomes burdensome and more prone to error. What are the work-arounds the authors propose, and what are the costs? How does that affect cost compared to other vendors that incorporate specimen and reagent recognition?

- Another limitation noted is the inability of the robot to detect specimen aspiration failures or clots, and presumably also short specimen volumes (a reality in the diagnostic laboratory). How do the authors propose this be addressed to minimize negative clinical impact of delayed or failed recognition of an invalid or (clinically)inaccurate test result?

Reviewer #2: 

The paper presents an affordable open-source robotic platform for SARS-CoV-2 testing by RT-qPCR using OT-2 open-source liquid-handling robots. The potential impact and need for this work is high. However, lack of rigor in each section of the paper diminishes the importance of this paper.

General comments to be addressed:

1. Abstract:

a. Bad grammar starting with the first sentence. The pandemic has struggled?? Please revise.

b. The abstract should summarize the need, methods, and results of the paper. After reading the paper it is very unclear what has been accomplished in the study. Please revise.

2. Introduction:

a. The advantages and disadvantages of different testing methods is not described.

b. The scale of testing is not well described.

c. Not clear what the contributions of this work is. A template and complete circuits??

d. Authors need to justify that the proposed method would provide an “optimal” solution or delete “optimal”.

3. Methods:

a. Very poor organization: It would be advised to start with a workflow picture describing the different elements of the proposed system. The section headers don’t follow a clear logic.

b. Not reproducible: The methods are not well enough described to reproduce your results. Showing your pseudo-code would help.

4. Results:

a. Poor organization

b. Figures need to be improved

c. Unclear study design and statistical evaluation

5. Conclusion is missing

6. General: The grammar and writing is very poor and needs to be improved.

6. PLOS authors have the option to publish the peer review history of their article (what does this mean?). If published, this will include your full peer review and any attached files.

Reviewer #1: **Yes: **Emilia Mia Sordillo, MD, PhD

Reviewer #2: No

---

## [Author Response · Author response to Decision Letter 0]

23 Apr 2021

Response to reviewers. Also available as .docx in the submission.

Reviewer #1: 

The title of the manuscript should be revised to be more informative, and, unless used with permission, to avoid the use of the name ROBOCOV, which is the name of a pre-existing statistical package – see https://github.com/kkdey/Robocov

R: We appreciate the insightful comment. Following the reviewer's suggestion we have removed the name ROBOCOV from the manuscript to avoid confusions and changed the title to make it more informative.

As noted by the authors, the COVID-19 pandemic has highlighted the disastrous impact of supply chain disruptions on global response. Identification of cost-effective alternatives to increase diagnostic capabilities is critical. The authors describe integration of a potentially interesting robotic platform from a relatively new vendor, that has the advantage of utilizing open-source software, and compare its performance to several better-known systems that have FDA emergency use authorization for SARS-CoV-2 diagnostic testing. However, the manuscript can be shortened and would benefit from being more focused. In addition, the use of advertising jargon, eg. "Hit the ground running", or descriptions of system advantages that are not referenced or specifically supported by data in the manuscript may appear promotional and detract from the authors’ message, and should be deleted.

R: We appreciate the succinct summary of our work and fully agree with the reviewer. Following these suggestions, we have shortened the manuscript and removed unnecessary parts and superfluous descriptions. We have also rewritten several parts of the manuscript to better reflect the objectives, the work we did, and its usefulness. The comments of the reviewer were particularly useful to focus the manuscript and enhance the discussion.

The authors’ stated aim is to describe use of the Opentrons OT system to set-up a reproducible workflow for diagnostic testing. It should be more clearly stated that the authors are describing a potential workflow incorporating the OT robot and the authors’ COVID-19 protocol for the Opentrons robot/author’s modifications to the COVID-19 protocol from the Opentrons Protocol Library, rather than a pilot study or an implementation in a clinical diagnostic setting, which would require testing of additional performance parameters and a larger sample size.

R: We apologize for the confusion and want to clarify our position. The CDB (Biomedical Diagnostic Center) works in a clinical setting (we are a part of the Hospital Clínic Barcelona). We have processed more than 40.000 samples (~500 runs) using the circuit described in the paper since the implementation of the platform. We do have other platforms/circuits in routine that we use in parallel.

The samples we use for comparing and validating the different platforms come from the EMQN (https://www.emqn.org/about-emqn/) which is the standard external quality assessment in Europe for molecular diagnostics.

We also want to stress that we have not used any protocol (code) available from the Opentrons Protocol Library, as we considered that the code was not good enough and difficult to generalize and/or re-adapt to changes. We have written all the necessary code from scratch, including several functions that enhance the current capabilities of the robots, as well as a template that is useful to build stations that conduct different steps from scratch. All the code has been made available through GitHub: https://github.com/CDB-coreBM/covid19clinic

We have included a setting section in Methods to better explain our context and the content we developed.

Finally, what we are validating is the adaptation of already validated kits (Thermo VP and VPII) into the OT-2 robotic platforms. 

Additionally, since the authors are proposing this system as an affordable system for clinical diagnosis when resources may be limited, it would be helpful to discuss further:

System Affordability

- When the authors describe the system as affordable relative to other systems, are they considering only the cost of the robot itself? If so, it should be clearly stated that other major set-up costs, maintenance, consumables, personnel costs for operation etc. needed to obtain an interpretable diagnostic result for clinical purposes have not been addressed or compared to other systems

R: We certainly agree with the reviewer that would be of the utmost interest to have a thorough comparison. But such comparison belongs in the field of health economics and would constitute an entire paper. Nonetheless, we have incorporated this limitation into the discussion section 

The OT-2 stations are a promising solution to increase the SARS-CoV-2 testing capability since it is an affordable open-source platform for liquid handling. The cost of the robot is much lower than other solutions, however other costs associated (personnel, consumables, etc) are similar. Other hospitals have compared the cost of using in-house protocols compared to commercial kits in OT-2 platforms12.

Consumables supply chain

- The authors raise concerns regarding global supply chain problems as a reason for using this open-source software platform. Please clarify why availability of the proprietary components and supplies listed on the Opentrons website, including heating blocks, pipette tips, racks & adapters, would not be an issue for this vendor.

R: We agree that supply chain problems should affect more or less uniformly the different vendors and Opentrons is clearly not an exception. What we tried to convey here:

Their openness confers them the ability to avoid limited supply chains for certain laboratory equipment, as it takes little time to define and test new laboratory equipment and include it in an already working protocol

 The clear advantage of this system is that it is very easy to substitute equivalent labware from different vendors interchangeably simply by modeling the components and defining it in the configuration files of the system. We have clarified this point in the paper. We actually only use the pipette tips from the company Opentrons (and have found a replacement if it was necessary); all the other labware was sourced from other vendors. In addition, racks, adapters, and other reusable parts can be obtained with 3D printing. We built some of them for some custom protocols, but as they were not included in this methodology they are not included in the manuscript.

Potential clinical impact of system limitations

- The authors note that there is no hardware or software in the robots to enable sample tracking. The value of robotics comes when large numbers of samples are being tested, and manual front and back end specimen tracking becomes burdensome and more prone to error. What are the work-arounds the authors propose, and what are the costs? How does that affect cost compared to other vendors that incorporate specimen and reagent recognition?

R: We thank the reviewer for the comment. The workaround that we propose is based on manually scanning each sample in the different numbered sample racks (1 to 4) and storing that information in a file that is passed automatically throughout the circuit across the different scripts. Therefore, a sample is always associated with a well position, that is the same in the different plates used. The plates are also tagged with the run name before the beginning of a process.

Additionally, other users are actively developing solutions to incorporate barcode scanning hardware which could allow this type of sample tracking with a minimal investment (100$): 

code: https://github.com/theosanderson/tube_checkout

demonstration: https://twitter.com/theosanderson/status/1286798512483708937

- Another limitation noted is the inability of the robot to detect specimen aspiration failures or clots, and presumably also short specimen volumes (a reality in the diagnostic laboratory). How do the authors propose this be addressed to minimize negative clinical impact of delayed or failed recognition of an invalid or (clinically)inaccurate test result?

R: This is clearly one of the main limitations. However, because the robots are modular and work in a chain, we have an assigned technician that prepares and supervises the whole run. He/she makes sure that there are no problems between stations by visually inspecting the plates and monitoring the behavior of the robots. 

Reviewer #2: 

The paper presents an affordable open-source robotic platform for SARS-CoV-2 testing by RT-qPCR using OT-2 open-source liquid-handling robots. The potential impact and need for this work is high. However, lack of rigor in each section of the paper diminishes the importance of this paper.

R: We thank the reviewer for the comments in our manuscript and agree that the potential impact of this work is high. Following the reviewer’s suggestions, we have included some new sections to better explain the objectives and logic behind our work. We believe that the code we have developed will not only be useful for SARS-CoV-2 testing, but will also facilitate the use of this platform in diverse research environments in the future.

General comments to be addressed:

1. Abstract:

a. Bad grammar starting with the first sentence. The pandemic has struggled?? Please revise.

R: We have thoroughly revised the manuscript for grammar errors and re-written most parts, including the abstract.

b. The abstract should summarize the need, methods, and results of the paper. After reading the paper it is very unclear what has been accomplished in the study. Please revise.

R: We have re-written the abstract and introduction and have included more specific objectives to clarify what has been accomplished and its usefulness.

2. Introduction:

a. The advantages and disadvantages of different testing methods is not described.

R: There is abundant literature comparing different testing methods. We feel that raising this point here would not add relevant context to our manuscript, as we already focus on the implementation of RT-qPCR.

b. The scale of testing is not well described.

R: Unfortunately, it is unclear to us what the reviewer is referring to with this comment. This could refer to the scale of testing needed globally (which is an open public health question), the number of samples we process in our implementation, or the theoretical capacity of a chain of such robots.

We have been using the platform regularly for a year approximately and have processed more than 40.000 samples so far. We did not run our setup at full capacity however, as we have access to several other robotic platforms.

As for the last question, one single chain of robots (4) can increment the throughput of a laboratory significantly. 

The complete process for testing 96 samples takes about 4h (Table 1), and a single laboratory technician can operate it. The longest step is the RT-qPCR amplification and detection using the ABI 7500 Fast thermocycler. Theoretically, a new run could start every 70 minutes; however, the inactivation of samples becomes the real bottleneck.

c. Not clear what the contributions of this work is. A template and complete circuits??

R: We agree that the contributions were not clear enough. We have re-written the introduction for clarity. We have indeed developed a framework that includes several functions and a proposed template (that includes all the functions) to construct stations from scratch. We also developed a circuit for SARS-CoV-2 RT-PCR testing with the framework developed.

d. Authors need to justify that the proposed method would provide an “optimal” solution or delete “optimal”.

We agree with the reviewer and have removed the word “optimal”

3. Methods:

a. Very poor organization: It would be advised to start with a workflow picture describing the different elements of the proposed system. The section headers don’t follow a clear logic.

We thank the reviewer for their comment. We agree that the organization was not clear enough, and we have re-ordered the manuscript to clarify the context and what we did. 

b. Not reproducible: The methods are not well enough described to reproduce your results. Showing your pseudo-code would help.

We have improved the code legibility on our GitHub repo and wiki (https://github.com/CDB-coreBM/covid19clinic/wiki ) with more detailed explanations of how the template is structured. We also include descriptions and examples of every individual function (included in the template) so it is easier to understand the logic behind them and its usefulness in different scenarios. 

We would like to stress that the code provided is fully reproducible (it has been implemented in other hospitals) when using the same labware and settings. However it can be fully customized once the user understands the code (hence the wiki page).

4. Results:

a. Poor organization

b. Figures need to be improved

c. Unclear study design and statistical evaluation

R: Thanks for the comments. We have re-organized the results so they follow a clear logic after explaining the methodology and moved figure 1 to supplementary material because we believe it distracted from the main message.

About the validation, the samples we use for comparing and validating the different platforms come from the EMQN (https://www.emqn.org/about-emqn/) which is the standard external quality assessment in Europe for molecular diagnostics. Specifically we use the Coronavirus Outbreak Preparedness EQA Pilot study. 

5. Conclusion is missing

R: We have improved the final message of the manuscript that acts as a conclusion.

6. General: The grammar and writing is very poor and needs to be improved.

R: As mentioned above we have revised the grammar and writing.

---

## [Decision Letter · Decision Letter 1]

18 May 2021

Implementation of an open-source robotic platform for SARS-CoV-2 testing by real-time RT-PCR

PONE-D-21-03093R1

Dear Dr. Villanueva-Cañas,

We’re pleased to inform you that your manuscript has been judged scientifically suitable for publication and will be formally accepted for publication once it meets all outstanding technical requirements.

Kind regards,

Ruslan Kalendar, PhD

Academic Editor

PLOS ONE

Additional Editor Comments (optional):

To authors, minor comment from reviewer:

Figure S1 (simulations for pipette height adjustment to ensure correct reagent aliquots ) referenced in the S1 File Supplementary methods appears to be missing from the materials available for review.

Reviewers' comments:

Reviewer's Responses to Questions

**Comments to the Author**

1. If the authors have adequately addressed your comments raised in a previous round of review and you feel that this manuscript is now acceptable for publication, you may indicate that here to bypass the “Comments to the Author” section, enter your conflict of interest statement in the “Confidential to Editor” section, and submit your "Accept" recommendation.

Reviewer #1: (No Response)

Reviewer #2: All comments have been addressed

2. Is the manuscript technically sound, and do the data support the conclusions?

Reviewer #1: Yes

Reviewer #2: Yes

3. Has the statistical analysis been performed appropriately and rigorously? 

Reviewer #1: (No Response)

Reviewer #2: Yes

4. Have the authors made all data underlying the findings in their manuscript fully available?

Reviewer #1: Yes

Reviewer #2: Yes

5. Is the manuscript presented in an intelligible fashion and written in standard English?

Reviewer #1: Yes

Reviewer #2: Yes

6. Review Comments to the Author

Reviewer #1: 

The authors have revised the manuscript and it is now more focused.

One concern - Figure S1 (simulations for pipette height adjustment to ensure correct reagent aliquots ) referenced in the S1 File Supplementary methods appears to be missing from the materials available for review

Reviewer #2: 

The paper presents an affordable open-source robotic platform for SARS-CoV-2 testing by RT-qPCR using OT-2 open-source liquid-handling robots. The potential impact and need for this work is high. The authors addressed all the previous comments and significantly improved the paper.

7. PLOS authors have the option to publish the peer review history of their article (what does this mean?). If published, this will include your full peer review and any attached files.

Reviewer #1: **Yes: **Emilia Mia Sordillo, MD, PhD

Reviewer #2: No

---

## [Editor Report · Acceptance letter]

30 Jun 2021

PONE-D-21-03093R1 

Implementation of an open-source robotic platform for SARS-CoV-2 testing by real-time RT-PCR. 

Dear Dr. Villanueva-Cañas:

I'm pleased to inform you that your manuscript has been deemed suitable for publication in PLOS ONE. Congratulations! Your manuscript is now with our production department. 

Kind regards, 

on behalf of

Prof. Ruslan Kalendar 

Academic Editor

PLOS ONE